# Effects of Dietary Rare Earth Chitosan Chelate on Performance, Egg Quality, Immune and Antioxidant Capacity, and Intestinal Digestive Enzyme Activity of Laying Hens

**DOI:** 10.3390/polym15071600

**Published:** 2023-03-23

**Authors:** Xinxin Lu, Xinyu Chang, Haijun Zhang, Jing Wang, Kai Qiu, Shugeng Wu

**Affiliations:** National Engineering Research Center of Biological Feed, Institute of Feed Research, Chinese Academy of Agricultural Sciences, Beijing 100081, China

**Keywords:** rare earth chitosan chelate, egg quality, digestive enzyme activity, immunity, antioxidant, laying hens

## Abstract

Rare earth chitosan chelate salt (RECC) is a potential feed additive and is a product of the chelation effect between rare earth ions and chitosan. This research study aims to explore the effects of dietary RECC on performance, egg quality, intestinal digestive function, and the immune and antioxidant capacity of laying hens in the late phase of production. A total of 360 56-week-old Dawu Jinfeng laying hens were randomly allotted into four treatment groups with six replicates per treatment and 15 birds per replicate. The laying hens were fed the basal diet supplemented with, respectively, 0 (control: CON), 100 (R1), 200 (R2), and 400 (R3) mg/kg for 8 weeks. Dietary RECC significantly improved average daily feed intake (ADFI) and average daily egg yield in both linear and quadratic manner (*p* < 0.05). In addition, albumen height and HU were improved significantly (*p* < 0.05) in a dose-dependent manner of RECC. In addition, a significant decrease (*p* < 0.05) in serum TP, IgA, and MDA for the R1 group and IgG in the R2 group were notable, while the increase in serum TP and decrease in T-AOC were found for R3 dietary group compared to CON (*p* < 0.05). The level of intestinal IL-2 and TNF-α was decreased by dietary RECC (*p* < 0.01). The activities of the digestive enzyme (α-Amylase, lipase, and Trypsin) showed a quadratic change with an increase and then decrease in response to increasing dose of RECC, 200 mg/kg RECC significantly increased the activity of lipase and Trypsin (*p* < 0.01). Supplementation of dietary RECC at low doses compared to higher doses impacted positive effects on the antioxidant capacity and immune function (*p* < 0.05). The utilization of RECC as a feed additive in the diet of aged laying hens exerted beneficial effects on egg production, albumen quality, humoral immunity, inflammatory response, and activity of digestive enzymes. Thus, the regulation of antioxidant capacity and duodenal function via increased enzyme activity and immune and inflammatory response were critical to the improvement of laying performance and egg quality in aged hens. The optimal supplemental dose is 100–200 mg/kg.

## 1. Introduction

Trace minerals, as constituents of hundreds of proteins, participate in the biochemical processes required for the normal growth and development of laying hens [1]. The absorption capacity of mineral elements decreases with increasing age during the late stages of the laying cycle of hens, resulting in decreased performance and egg quality [2]. The inclusion of organic mineral elements in the diet is a common nutritional regulation strategy to improve the immune function and antioxidant capacity of laying hens [3,4].

Rare earth element is a collective term used to describe a set of seventeen metallic elements, including scandium (Sc), yttrium (Y), and 15 kinds of lanthanides on the periodic table, and they are mainly found in crustal ores [5]. Organic and inorganic rare earth elements (REEs) have been widely used as safe dietary substitutes for antibiotics to improve the health status and performance of farm animals, including cattle, sheep, rabbits, pigs, and laying hens [6,7,8]. Specific to laying hens, besides enhancing the antioxidant capacity and egg production, REEs increases the activity of intestinal enzymes and the concentration of calcium and phosphorus in the blood, which in turn improve feed efficiency and eggshell quality [9,10,11]. Chitosan is the world’s second most abundant renewable biopolymer, could remove free radicals in the intracorporeal, elevate the antioxidant and immunity capacity of the organism, and positively impact livestock production as a feed additive [12,13]. Supplementation of dietary chitosan induced changes in serum metabolites involved in muscle metabolism, which improved the growth performance and meat quality of lambs [14]. Low-molecular-weight chitosan ameliorates the intestinal health of weanling pigs [15]. More importantly, chitosan molecules contain a large amount of hydroxyl and amino groups, which are good ligands for metal ions. Chitosan chelating metal ions could improve the biocompatibility of metal ions and the absorption and utilization rate in vivo, which may account for the improved antioxidant and antibacterial activities of chitosan [16]. Rare earth chitosan chelate salt (RECC), a new chelate salt, was produced through the coordination chelation effect between rare earth ions and chitosan [17]. Dietary supplementation with 175~200 mg/kg of RECC improved the immune and anti-oxidant capacity and growth performance of broilers [18], 200 mg/kg RECC improved the growth, immunity, and microbial balance of weaned piglets [19], 800 mg/kg enhanced the growth performance and immunity of *Carassius auratus gibelio* [20], and 5 g/kg may improve intestinal health and digestive capacity of *Leiothrix lutea* [21]. Therefore, RECC chelated through special processing with chitosan probably has both the chemical properties of rare earth and the biological function of chitosan.

The later stage of the laying period is the key period of nutrition regulation for laying hens. This study will explore the effect of RECC as a feed additive on the physiological response, laying performance, and egg quality of aged laying hens. Through biochemical analysis, ultraviolet spectrophotometry, enzyme-linked immunosorbent assay, and other molecular experiments to detect the metabolic indicators of the organism reveal the impact path from the perspective of the organism’s antioxidant and immune capacity and intestinal digestive enzyme activity. This study will provide a reference for the appropriate inclusion to improve economic benefits.

## 2. Materials and Methods

### 2.1. Animals and Experimental Diets

The experiment adopted a completely randomized design. A total of 360 healthy 56-week-old DawuJinfeng laying hens with similar body weight (BW) were selected and randomly assigned to 1 of the treatments with 6 replicates per treatment and 15 birds per replicate. Three birds were raised in one cage (40 × 40 × 37 cm) with 16 h/d at 20 Lx and relative humidity of 50~90% in a naturally ventilated house with a temperature under 30 °C. A two-week adjusting period was spent for birds to adapt to the basal diet before the 8-week trial. The laying hens were given a routine immunization and free access to feed and water. The dead and sick birds were removed and recorded immediately. Total egg number, abnormal egg number, and total egg weight per replicate were recorded every day, and feed intake was recorded every week. The feed conversion ratio (FCR) was obtained by feed intake/egg weight (g/g).

The basal diet (Mash form) was formulated according to the NRC (1994) and NY/T 33-2004 standards, and its composition and nutrient content are shown in Table 1. The control group (CON) is the basal diet. Three experiment groups (R1, R2, and R3) consist of the basal diet supplemented with 100, 200, and 400 mg/kg of RECC, respectively. RECC used in the present experiment was produced by the Institute of Feed Research of the Chinese Academy of Agricultural Sciences.

### 2.2. Sample Collection

At the end of weeks 4 and 8, 5 normal eggs per replicate were collected for the determination of egg quality. At the end of the trial, one bird per replicate was selected and fasted for more than 8 h. Blood samples were collected from the wing vein into tubes, placed in a water bath at 37 °C for 1 h, and centrifuged at 3000 r/min for 10 min. The harvested serum samples were transferred to 1.5 mL Eppendorf tubes and stored at −20 °C. Subsequently, the birds were slaughtered, and the intestinal tract was removed and placed on ice. Approximately 1.5 cm of the duodenum, jejunum, and ileum were gently cut off and fixed in the 4% formaldehyde buffer. The duodenal contents and jejunal mucosa were collected and quick-frozen in liquid nitrogen and then stored at −80 °C.

### 2.3. Egg Quality Determination

The egg shape was measured using an egg-shape index apparatus (Egg Index Reader, Fujihira Industry Co., Tokyo, Japan). Albumen height, Haugh unit, and yolk color were measured using the egg quality auto-analyzer (Egg Analyzer ™, Orka Technology Ltd., Ramat Hasharon, Israel). Eggshell strength was measured using an eggshell strength analyzer (Egg Force Reader ™, EFR-01, Orka Food Technology Ltd., Ramat Hasharon, Israel). Eggshell thickness based on three points (the air cell, equator, and sharp end) was determined using Egg Shell Thickness Gauge (Orka Technology Ltd., Ramat Hasharon, Israel).

### 2.4. Chemical Analysis

The levels of aspartate aminotransferase (AST), total protein (TP), albumin (ALB), blood glucose (Glu), uric acid (UA), and triglycerides (TG) in serum were measured using kits purchased from Shanghai Kehua Biological Engineering Co., Ltd. (Shanghai, China). The total antioxidant capacity (T-AOC), superoxide dismutase (SOD) activity, malondialdehyde (MDA) content, glutathione (GSH) concentration, and the glutathione peroxidase (GSH-Px) activity in serum were measured by ABTS, WST-1, TBA, and colorimetric assay, respectively. The kits were purchased from Nanjing Jiancheng Bioengineering Co., Ltd. (Nanjing, China).

The activities of trypsin, amylase, and lipase in the duodenal contents were measured using commercial kits. The trypsin test kit, α- Amylase AMS assay kit, and lipase LPS assay kit were purchased from Nanjing Jiancheng Bioengineering Co., Ltd. (Nanjing, China). The jejunal mucosa samples were homogenized well with a certain amount of PBS (pH 7.4) and then centrifuged at 3000 r/min for 20 min. The supernatant obtained was used to measure the levels of secreted immunoglobulin A (sIgA), interleukin-2 (IL-2), interleukin-6 (IL-6), and tumor necrosis factor-α (TNF-α). The assays were performed using kits purchased from Shanghai enzyme-linked Biotechnology Co., Ltd. (Shanghai, China). All of the above procedures were conducted based on strict adherence to the manufacturer’s instructions.

Information regarding the commercial kits is listed in Supplemental Appendix A.

### 2.5. Statistical Analysis

All data were processed by one-way ANOVA procedure in SPSS Version 19.0 software (SPSS Inc., Chicago, IL, USA). Duncan’s multiple comparisons were conducted to assess the differences between groups, and the Contrast command was performed to assess the linear and quadratic effects of RECC. Significant differences between groups are considered at *p* < 0.05, and a significant trend is considered at 0.05 ≤ *p* < 0.10.

## 3. Results

### 3.1. Performance

The effects of dietary RECC addition on the growth performance in late-laying hens are shown in Table 2. The results obtained from one-way ANOVA analysis showed that dietary treatments did not affect (*p* > 0.05) the average daily egg yield, average egg weight, and FCR of hens during the trial period. However, average daily egg yield and ADFI increased linearly and quadratic with the increase in dietary RECC supplementation (*p* < 0.05), and the average egg weight tended to increase linearly (*p* < 0.1). ADFI of hens fed with RECC was significantly higher (*p* < 0.01) than CON, especially at the terminal stage of the experiment (weeks 7~8). Specifically, in R1, R2, and R3 groups increased by 4.42%, 6.36%, and 5.87%, respectively.

### 3.2. Egg Quality

As shown in Table 3, Albumen height and HU at week 4 were linearly increased by RECC supplementation in diets (*p* < 0.05). No significant differences were observed in egg weight, eggshell strength, eggshell thickness, egg shape, HU, and albumen height between every treatment at weeks 4 and 8 (*p* > 0.05).

### 3.3. Serum Indexes

The serum biochemical indices of the laying hens are shown in Table 4. There was no significant effect (*p* > 0.05) of dietary RECC on the activity of GSH-Px and concentration of AST, GLU, UA, TG, GSH, SOD, and IgM in the serum.

The concentrations of TP and ALB in serum were affected by dietary supplementation with RECC. They showed a quadratic correlation with the rise of RECC content that first decreased and then increased (*p* < 0.05). The TP value of R1 was lower than that of CON, with 45.42 g/L, and that of R3 was higher than CON, with a value of 55.17 g/L (*p* < 0.01). Similarly, the same trend was observable for ALB, where the minimum value was 13.22 g/L in R1, and the maximum value was 14.87 g/L in R3 (*p* < 0.05).

Dietary RECC has a significant effect on serum immunoglobulins. IgA level in serum was significantly lower in R1 than in CON (*p* < 0.05). Serum IgG level in R2 was significantly lower than that in CON and R1 (*p* < 0.05) and showed a linearly decreasing trend (*p* < 0.10).

The values of MDA and T-AOC were significantly affected by RECC (*p* < 0.05). The content of serum MDA in R1 was significantly lower than in CON and R3, presenting a quadratic change with a decrease and then an increase as dietary RECC increased (*p* < 0.05). The serum T-AOC in R3 was significantly lower than in CON and showed a variation rule similar to MDA (*p* < 0.01).

### 3.4. Intestinal Immune

As shown in Table 5, no significant differences due to dietary treatments for IL-6 and sIgA levels in the intestinal mucosa (*p* > 0.05) were found. IL-2 level was significantly lower in all RECC groups compared to CON and gradually decreased with the increase in RECC addition (*p* < 0.01). Dietary RECC reduced the TNF level in R1 and R3 (*p* < 0.01) but exerted no effect on the same function for the R2 group.

### 3.5. Digestive Enzyme Activity

As shown in Table 6, supplementation of dietary RECC at varying inclusion levels had significant effects on the activities of AMS (*p* < 0.05) and a highly significant influence on LPS and Trypsin (*p* < 0.01). Specifically, the R2 group recorded higher significant values compared to other treatments. The activities of AMS, LPS, and trypsin showed quadratic change with an increase and then a decrease (*p* < 0.05) in response to RECC in a dose-dependent manner. Among them, LPS and Trypsin changes were particularly remarkable (*p* < 0.01). Supplementation of RECC at 200 mg/kg in the diet recorded the highest enzymatic activity in the duodenum of laying hens.

## 4. Discussion

Plenty of studies suggest that dietary supplementation with RECC increases the average daily egg yield [22], promoting the weight gain of different animals, broilers [18], pigs [23], rabbits [24], fish [20], etc. In the current study, the appropriate RECC level in diets was also propitious for laying performance, but the improvement of RECC on FCR in previous studies was not reconfirmed here, which is consistent with the results of Li et al. [25]. This may be because dietary RECC augmented the digestive enzyme activity in the intestinal tract, which invariably enhanced the bioavailability of nutrients, absorption, and utilization, while the duration for the feeding trial was not sufficient enough to induce changes that were reflected in the FCR of the laying hens.

REEs, as a natural antioxidant, have similar physical and chemical properties to calcium, which is of beneficial impact on egg quality [26]. Similarly, supplementation of 200 mg/kg of cerium oxide (CeO) and lanthanum oxide (LaO) in the diet of laying hens improved the eggshell strength, albumen quality, and HU [10,11,22]. By far, the effects of chitosan on egg quality are met with inconsistent results. It could improve albumen height, HU, egg weight, eggshell strength, and yolk color [27,28,29] or not affect egg quality [7,30]. In the present study, albumen quality increased linearly with the inclusion dosage of RECC. It could be deduced that RECC is a compound consisting of REEs and chitosan, which has the potential to improve egg quality.

Chitosan and its derivatives are excellent antioxidants [31,32,33,34] and could modulate oxidative stress by improving intestinal barrier function [35]. Varying inclusion levels of REEs have been found to exert different impacts on their oxidative activity [36,37,38]. Recent research shows that REE supplementation increased the activities of SOD and CAT and decreased MDA, thereby reducing the reserves of cellular reactive oxygen species, preventing oxidative-induced cell damage, and improving antioxidant capacity [39,40]. However, the conclusions of Durmuş and Bölükbaşı are contradictory; although dietary supplementation of CeO and LaO decreased MDA content, the activity of SOD was also reduced in the plasma [10,11]. Nevertheless, until now, there has been relatively little research on the effect of RECC on the antioxidant capacity of laying hens. Only recent Cheng et al. research on broilers pointed out that rare earth-chitosan chelate (RECC) improved the antioxidant capacity of blood and liver [18]. In the current study, dietary supplementation with RECC reduced the content of MDA and the activity of T-AOC in serum, which was similar to previous reports [10,11,41]. The SOD activity in serum showed a downward trend at a low dosage of dietary RECC inclusion. Similarly, the same effects were observable when REEs was supplemented in the diets of fish or poultry species [37,42,43]. It may be due to the multiple effects of the antioxidant enzymes (CAT, GSH-Px, etc.) and non-enzymatic defense (fat-soluble vitamins and water-soluble vitamins) [37]. In the current study, supplementation of dietary RECC at an inclusion level of 100 mg/kg was beneficial to the antioxidant capacity of laying hens, and at an inclusion level of 400 mg/kg, no adverse effect was notable in the laying hens. The beneficial antioxidant capacity stems from the capacity of RECC to reduce lipid peroxidation and oxidative stress induced cell damage. It is consistent with the report that RECC improved serum and liver antioxidant capacity in broilers [18]. In the present study, the improvement in the antioxidant status of laying hens is probably due to the capacity of RECC to stimulate the non-enzymatic antioxidant system instead of the enzymatic antioxidant system, which is consistent with the previous study on chitosan [29].

The blood protein content is closely related to the immune function of animals. During pathogen invasion, it promotes inflammation and complement cascades and binds to specific antigens in the form of immunoglobulins. Thus, its surplus or deficient concentration in the blood depicts a damaged immune mechanism or a sub-health status [44]. Alterations in liver functions or hepatic cell damage often induce an increase in the concentration of ALT and AST in the serum of animals [45]. The normal range of TP concentration in the serum of birds is 35~55 g/L [46]. Serum TP content in the present study was within the normal range for chickens, and no significant change was observed for serum AST. Birds fed diets supplemented with 0~400 mg/kg of RECC showed normal hepatic function, unimpaired protein synthesis, and stable health status. When 400 mg/kg of RECC was added to the diet, the highest value (55.17 g/L) for serum TP level was recorded. It could be speculated that supplementation of RECC at an inclusion level exceeding 400 mg/kg may induce physiological dysfunctions, including abnormal protein metabolism, damaged liver function, and triggered immune and inflammatory reactions, which all harm the birds healthwise. Serum ALB, a protein reservoir, is the most favorable source of amino acids for synthesizing tissue protein, and its content has a close correlation with the TP concentration [20,47]. This study confirmed this viewpoint that ALB serum has the same change as TP in response to dietary RECC.

Immunoglobulins (Ig), produced by lymphocytes and plasma cells, exert protective effects such as anti-inflammatory, anti-disease, and anti-infection immunity on the organism [48]. In the present study, IgA and IgG levels in serum decreased and then increased along with the linear increase in RECC supplementation, which suggests that RECC could regulate humoral immunity and affect the immune function of laying hens. However, it is inconsistent with previous studies as follows. The addition of chitosan increased the content of IgA, IgG, and IgM in the serum of Huoyan geese, thereby improving immunity [49]. Supplementing chitooligosaccharides to the diet of laying hens does not affect serum concentrations of IgA, IgG, and IgM [29]. To date, the effects of REEs on the immune response have not been extensively studied [6]. Azomite enriched with REEs increased the levels of IgG and IgA in broilers [50]. Rare earth element lanthanum (REE-La) has double influences on the serum Ig level. Long-term exposure to a low oral dose of REE-Y^3+^ could markedly elevate the levels of IgM and IgG in serum, while high-dose exposure tends to lower the Ig levels [51]. However, the serum IgM level was significantly decreased by the administration of 20 mg/kg of lanthanoids, and the immunity and liver function was disturbed in mice [52]. Therefore, an inhibitory or stimulator influence prompted by RECC on immunoregulation relies on the inclusion level/ dosage. The mechanism by which RECC changes immunophenotyping needs further research.

IgA could activate the immune function by modulating the production of key cytokines such as TNF, IL-6, and IL-2 by human myeloma cells [53]. Rare earth-chitosan chelate caused a linear decrease in serum IL-1b and IL-6 and a significant increase in IL-2 and IL-4 levels in 21-day-old broilers, and the reduced content of intestinal proinflammatory factors was beneficial to the body’s immune status [18]. Chitosan oligosaccharide can activate rat frontal macrophages, promote the release of intestinal inflammatory factors IL-6, TNF-a, IFN-1, and IL-10, show resistance to endotoxin, promote the recovery of tissues, and suppress inflammation [54,55]. Broilers with 100 mg/kg COS had higher serum concentrations of IL-1β, IL-6, TNF-α, IgM, and IFN-γ, optimizing macrophage function and improving immune function [56]. However, in rats exposed to cerium oxide nanoparticles, the ingested CeO2 activated alveolar macrophages and lymphocytes to secrete proinflammatory cytokines IL-12 and IFN-γ, which induced an inflammatory response [57]. In the current study, dietary RECC at 100 mg/kg inclusion level reduced the levels of intestinal cytokines, including IL-2 and TNF-α. The reduced level of proinflammatory cytokines would mean an improvement in the health status. In addition, it indicated that RECC could improve the immune function of intestinal mucosa of laying hens in a dose-dependent manner. Thus, it could be that the effect of REEs on animals is dependent on the form and inclusion level. However, the effects of dietary RECC on the immune system of poultry need to be further explored.

The small intestine is important for the digestion and absorption of nutrients, and the increase in the content and activity of digestive enzymes is conducive to enhancing the ability of absorption. This study would be the first to provide insight into the relationship between dietary RECC and digestive enzyme activity in laying hens. With the increase in RECC supplementation, the activities of digestive enzymes appeared to follow a quadratic change with an increase and then a decrease. The inclusion of dietary RECC at the dosage of 200 mg/kg exerted a significant effect on digestive enzyme activity. Compared with the treatment without special disposal, supplementation of 200 mg/kg of RECC in the diet improved the activities of amylase, lipase, and trypsin in the duodenum, whereas adding 400 mg/kg RECC diminished the amylase activity. Similar results to this research were obtained in both test trials of feeding chitosan to *Paramisgurnus dabryanus* [58] or REEs mixture to carp [59]. Chitosan enhances the secretion of digestive enzymes, probably due to its ability to increase the proliferation of beneficial microbes and suppress pathogen colonization in the gut [21]. The absorption of REEs is positively relevant to their accumulation in the gastrointestinal tract, which further inhibits the growth of harmful bacteria, stimulates the production of antibacterial and anti-inflammatory factors, improves the activities of digestive enzymes, and strengthens digestion and absorption of nutrients [8]. In this research, the addition of RECC to the diet at an appropriate inclusion level enhanced the activities of amylase, lipase, and trypsin in the duodenum, and the effect was higher than that of individually applied chitosan or REEs in diets. REEs have similar characteristics to Ca^2+^, could accelerate self-digestion and activation of trypsinogen, and promote protease activity [21]. In addition, the oligosaccharide formed by the digestion of chitosan played an auxiliary role in promoting the absorption of Ca^2+^ [60]. Therefore, the underlying mechanism may be attributed to the fact that trypsinogen transforms into trypsin under the catalysis of REEs, and REEs prevents trypsin from self-digestion. One review report on REEs concluded that the capacities of enzymes in the digestive system were influenced by dietary supplementation with REEs or Re-based compounds [61]. In future studies, the mechanism of RECC acting on intestinal digestive enzyme activities is worth exploring further.

## 5. Conclusions

The utilization of RECC as a safe feed additive in laying hens has been proven once again. Dietary RECC improved egg production and egg quality in laying hens at the late phase of egg production. The beneficial effects on egg performance are adducible to the positive effects of RECC on the antioxidant capacity, anti-inflammatory effect, increasing humoral immunity, and enhancing intestinal digestive enzymes activity. Supplementation of RECC at an inclusion level of 100~200 mg/kg provided an optimal performance and physiological response in the aged laying hens.

## Figures and Tables

**Table 1 polymers-15-01600-t001:** Composition and nutrient levels of the basal diet of laying hens (as-feed basis).

Items	Content
Ingredient (%)	
Corn	62.20
Soybean meal	26.98
Soybean oil	0.50
Limestone powder	8.50
CaHPO_4_	0.90
NaSO_4_	0.25
Salt	0.15
Choline chloride	0.12
Vitamin premix ^1^	0.03
Mineral premix ^1^	0.10
DL-methionine (DL-Met)	0.12
Phytases	0.05
Zeolite powder	0.10
Total	100.00
Nutrients ^2^	
ME (Mcal/kg)	2.70 (2.65)
Crude protein (CP) (%)	16.70 (16.50)
Calcium (Ca) (%)	3.37 (3.50)
Available phosphorus (AP) (%)	0.25 (0.32)
Lysine (Lys) (%)	0.85 (0.75)
Methionine (Met) (%)	0.38 (0.34)
Met + Cysteine (Cys) (%)	0.64 (0.65)

^1^ Vitamin and mineral premix (per kg diet provided): Cu, 8 mg (CuSO_4_·5H_2_O); Fe, 80 mg (FeSO_4_); Zn, 88 mg (ZnSO_4_·H_2_O); Se, 0.3 mg (Na_2_SeO_3_); I, 0.7 mg (KI); Vitamin A, 8800 IU (retinyl acetate); Vitamin D_3_ 3300 IU; VE, 16.9 IU; vitamin K3, 2.25 mg; VB1, 1.7 mg; VB2, 5.5 mg; VB6, 3.3 mg; VB12, 0.022 mg; Biotin, 0.08 mg; Folic acid, 0.71 mg; Pantothenic acid, 6.9 mg; Niacinamide, 28 mg. ^2^ The nutrition level is the calculated value. The values in the parentheses are the standards of nutritional requirement referring to NRC (1994) and NY/T 33-2004.

**Table 2 polymers-15-01600-t002:** Effects of dietary rare earth chitosan chelates (RECC) on the performance of laying hens.

Items	Dietary RECC Level (mg/kg)				SEM	*p*-Value		
	0	100	200	400		A	L	Q
Average daily egg yield (%)								
1~2 wk	82.86	83.89	84.92	88.48	0.99	0.245	0.039	0.119
3~4 wk	85.56	87.15	86.72	89.68	1.08	0.615	0.197	0.436
5~6 wk	84.80	82.48	88.30	87.32	0.98	0.140	0.155	0.361
7~8 wk	81.59	83.94	89.45	90.10	1.43	0.084	0.019	0.044
1~4 wk	84.21	85.52	85.82	90.14	0.98	0.192	0.032	0.094
5~8 wk	83.19	83.21	88.87	88.02	1.09	0.113	0.018	0.055
1~8 wk	83.70	84.36	87.35	87.95	0.96	0.316	0.011	0.040
ADFI (g)								
1~2 wk	108.65	107.09	107.68	107.53	0.51	0.726	0.608	0.712
3~4 wk	103.04	106.04	102.85	103.23	1.25	0.802	0.806	0.930
5~6 wk	109.48	107.98	109.35	109.75	0.83	0.894	0.737	0.878
7~8 wk	105.43 ^b^	110.09 ^a^	112.14 ^a^	111.62 ^a^	0.80	0.005	0.008	0.001
1~4 wk	105.85	106.57	105.27	105.38	0.71	0.927	0.690	0.925
5~8 wk	106.26 ^b^	109.04 ^ab^	110.75 ^a^	110.69 ^a^	0.62	0.032	0.016	0.011
1~8 wk	106.65	108.71	108.01	108.03	0.39	0.320	0.379	0.331
FCR (g/g)								
1~2 wk	2.17	2.14	2.13	2.12	0.03	0.930	0.526	0.799
3~4 wk	1.98	2.02	1.97	1.92	0.02	0.492	0.222	0.374
5~6 wk	2.13	2.18	2.05	2.12	0.03	0.624	0.719	0.830
7~8 wk	2.12	2.16	2.05	2.08	0.03	0.660	0.467	0.754
1~4 wk	2.08	2.08	2.05	2.02	0.02	0.750	0.284	0.559
5~8 wk	2.13	2.17	2.05	2.10	0.03	0.535	0.557	0.771
1~8 wk	2.10	2.13	2.05	2.06	0.02	0.604	0.352	0.644
Average egg weight (g)								
1~2 wk	60.68	59.85	60.00	59.49	0.24	0.383	0.123	0.279
3~4 wk	61.00	60.39	60.63	60.00	0.22	0.449	0.147	0.357
5~6 wk	61.22	60.85	60.79	60.19	0.20	0.366	0.073	0.209
7~8 wk	61.61	61.49	61.06	60.67	0.20	0.332	0.062	0.183
1~4 wk	60.84	60.12	60.31	59.75	0.22	0.373	0.114	0.283
5~8 wk	61.42	61.17	60.93	60.43	0.19	0.329	0.058	0.172
1~8 wk	61.13	60.64	60.62	60.09	0.20	0.352	0.074	0.208

*p*-value: A represents one-way ANOVA and Duncan’s multiple comparisons; L and Q represent linear and quadratic analysis using regression analysis, respectively. ^a,b^ Means within a row with no common superscripts differ significantly (*p* < 0.05). ADFI, average daily feed intake; FCR, feed conversion ratio (feed:egg, g:g); SEM, standard error of means.

**Table 3 polymers-15-01600-t003:** Effects of dietary rare earth chitosan chelates (RECC) on egg quality of laying hens.

Items	Dietary RECC Level (mg/kg)				SEM	*p*-Value		
	0	100	200	400		A	L	Q
Week 4								
Egg weight (g)	62.31	61.71	61.04	61.60	0.27	0.454	0.375	0.280
Eggshell Strength (N/m^2^)	41.18	38.24	39.01	38.49	0.63	0.347	0.244	0.312
Eggshell Thickness (mm)	0.453	0.447	0.444	0.440	0.003	0.556	0.160	0.348
Egg shape index	1.358	1.368	1.367	1.363	0.003	0.745	0.772	0.601
Haught unit	70.64	71.11	73.48	73.84	0.60	0.124	0.026	0.076
Albumen Height (mm)	5.48	5.50	5.72	5.80	0.06	0.186	0.036	0.113
Week 8								
Egg weight (g)	61.75	62.37	61.42	61.96	0.32	0.790	0.712	0.935
Eggshell Strength (N/m^2^)	35.07	36.85	37.85	35.08	0.56	0.208	0.858	0.099
Eggshell Thickness (mm)	0.430	0.443	0.442	0.432	0.003	0.439	0.437	0.431
Egg shape index	1.356	1.359	1.364	1.355	0.003	0.765	0.890	0.598
Haught unit	77.12	75.16	77.71	76.57	0.54	0.398	0.912	0.994
Albumen Height (mm)	6.30	6.36	6.35	6.24	0.06	0.883	0.630	0.716

*p*-value: A represents one-way ANOVA and Duncan’s multiple comparisons; L and Q represent linear and quadratic analysis using regression analysis, respectively. SEM, standard error of means.

**Table 4 polymers-15-01600-t004:** Effects of dietary rare earth chitosan chelates (RECC) on serum indexes of laying hens.

Items	Dietary RECC Level (mg/kg)				SEM	*p*-Value		
	0	100	200	400		A	L	Q
AST (U/L)	185.67	178.00	175.67	196.17	4.46	0.379	0.320	0.207
TP (g/L)	50.42 ^b^	45.42 ^c^	52.77 ^ab^	55.17 ^a^	0.98	<0.001	0.004	0.009
ALB (g/L)	14.05 ^ab^	13.22 ^b^	13.90 ^b^	14.87 ^a^	0.19	0.014	0.039	0.014
GLU (nM)	8.40	9.12	8.23	9.43	0.25	0.264	0.258	0.439
UA (U/L)	158.20	144.40	141.50	181.25	8.56	0.412	0.355	0.228
TG (mM)	31.64	28.02	31.37	29.55	0.70	0.967	0.900	0.989
IgA (g/L)	0.311 ^a^	0.262 ^b^	0.290 ^ab^	0.291 ^ab^	0.01	0.035	0.734	0.220
IgG (g/L)	2.11 ^a^	2.13 ^a^	1.94 ^b^	2.01 ^ab^	0.03	0.029	0.062	0.099
IgM (g/L)	1.025	0.982	1.050	1.064	0.02	0.437	0.352	0.652
MDA (mM)	5.48 ^a^	2.92 ^b^	3.92 ^ab^	5.36 ^a^	0.38	0.030	0.543	0.042
GSH (mg/L)	17.32	19.88	7.71	10.23	2.87	0.412	0.252	0.462
GSH-Px (U/mL)	3053	2568	3167	3270	179	0.586	0.438	0.683
SOD (U/mL)	85.03	74.40	83.38	84.19	1.70	0.099	0.637	0.420
T-AOC (mM)	0.734 ^a^	0.723 ^a^	0.653 ^ab^	0.587 ^b^	0.02	0.010	0.001	0.004

*p*-value: A represents one-way ANOVA and Duncan’s multiple comparisons; L and Q represent linear and quadratic analysis using regression analysis, respectively. ^a,b,c^ Means within a row with no common superscripts differ significantly (*p* < 0.05). AST, aspartate aminotransferase; TP, total protein; ALB, Albumin; GLU, glucose; UA, Urea; TG, triglyceride; IgA, immunoglobulin A; IgG, immunoglobulin G; IgM, immunoglobulin M; MDA, malondialdehyde; GSH, glutathione; GSH-Px, glutathione peroxidase; SOD, superoxide dismutase; T-AOC, total antioxidant capacity; SEM, standard error of means.

**Table 5 polymers-15-01600-t005:** Effects of dietary rare earth chitosan chelates (RECC) on intestinal immune indexes of laying hens.

Items	Dietary RECC Level (mg/kg)				SEM	*p*-Value		
	0	100	200	400		A	L	Q
IL-2 (ng/mL)	3.857 ^a^	3.419 ^b^	3.316 ^bc^	3.064 ^c^	0.08	<0.001	<0.001	<0.001
IL-6 (ng/mL)	0.532	0.517	0.559	0.535	0.01	0.303	0.601	0.684
TNF-α (ng/mL)	1.353 ^a^	1.168 ^b^	1.326 ^a^	1.207 ^b^	0.02	0.001	0.132	0.302
sIgA (μg/mL)	22.30	22.50	23.22	21.95	0.28	0.437	0.669	0.329

*p*-value: A represents one-way ANOVA and Duncan’s multiple comparisons; L and Q represent linear and quadratic analysis using regression analysis, respectively. ^a,b,c^ Means within a row with no common superscripts differ significantly (*p* < 0.05). IL-2, interleukin 2; IL-6, interleukin 6; TNF-α, tumor necrosis factor-α; sIgA, secretory immunoglobulin A; SEM, standard error of means.

**Table 6 polymers-15-01600-t006:** Effects of dietary rare earth chitosan chelates (RECC) on duodenal digestive enzyme activity of laying hens.

Items	Dietary RECC Level (mg/kg)				SEM	*p*-Value		
	0	100	200	400		A	L	Q
AMS (U/mg prot)	437.77 ^ab^	459.14 ^ab^	551.26 ^a^	296.23 ^b^	33.34	0.036	0.116	0.022
LPS (U/g prot)	124.04 ^b^	173.28 ^b^	274.47 ^a^	155.38 ^b^	18.02	0.010	0.517	0.010
Trypsin (U/mg prot)	284.68 ^b^	371.62 ^b^	571.44 ^a^	339.55 ^b^	33.93	0.008	0.529	0.009

*p*-value: A represents one-way ANOVA and Duncan’s multiple comparisons; L and Q represent linear and quadratic analysis using regression analysis, respectively. ^a,b^ Means within a row with no common superscripts differ significantly (*p* < 0.05). AMS, α-amylase; LPS, lipase; SEM, standard error of means.

## Data Availability

The raw data of this article will be available without reservation by contacting the corresponding author.

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
