# Peer review of "Effects of Dietary Rare Earth Chitosan Chelate on Performance, Egg Quality, Immune and Antioxidant Capacity, and Intestinal Digestive Enzyme Activity of Laying Hens"

_polymers, 2023, doi:10.3390/polym15071600_

Round 1

Reviewer 1 Report

This article is comprehensive, logically organized, and contains valuable information on the effects of dietary rare earth chitosan chelate (RECC) salt on performance, egg quality, immune and antioxidant capacity, and intestinal digestive enzyme activity of laying hens. The authors did excellent research to explore the effects of dietary supplementation with RECC salt on performance, egg quality, intestinal digestive function, and immune, and antioxidant capacity of laying hens in the late phase of production. The authors demonstrated that the appropriate RECC salt in diets had positive effects on average daily feed intake, egg yield, and egg quality of the aged laying hens, which was realized probably through enhancing serum antioxidant capacity to relieve organism oxidative stress, improving immune function in serum and intestinal mucosa to reduce inflammation reaction, and stimulating digestive enzyme activity to strengthen the duodenal function. The authors presented the diet composition and nutrient levels of laying hens (air-dried basis) in Table 1, however, the authors should present the standard deviations of the diet composition and nutrient levels for the readability and reliability of those data. The submitted manuscript has significant scientific insights and the conclusions are soundly supported by the experimental data. However, the present submission requires minor revisions before being considered for publication in the Special Issue: Natural and Synthetic Polymers-Hopes and Fears in an Era of Ecological Modernization in its current condition.

Reviewer 2 Report

Dear Editor and Authors, thank you for giving me the opportunity to review this manuscript entitled “Effects of Dietary Rare Earth Chitosan Chelate on Performance, egg Quality, Immune and Antioxidant Capacity, and Intestinal Digestive Enzyme Activity of Laying Hens" I have read the manuscript, and I believed it was an interesting study.

However, the manuscript needs major revisions to consider for publication.

My Specific Comments: 

1. # Written English and grammatical mistakes need to be corrected.  A person proficient in written English edits the manuscript.

2. Abstract: Well written but can be improved with the incorporation of total results and conclusions.

3. Generally, the introduction should include more information on methodologies and detecting techniques and suggested to include more recent references in place of old ones

4. Overarching goals of your study must be concluded in the introduction?

5. Figures must be included in this paper.

6. The discussion and comparison with relevant international studies are limited

7. Conclusions: rewrite this section with more appropriate information with clarity.

Round 2

Reviewer 2 Report

I am satisfied with the author's responses to my questions